# Low Preoperative Cachexia Index Is Associated with Severe Postoperative Morbidity in Patients Undergoing Gastrectomy for Gastric Cancer

**DOI:** 10.3390/diagnostics15182284

**Published:** 2025-09-09

**Authors:** Melih Can Gül, Muhammet Kadri Çolakoğlu, Volkan Öter, Neslihan Karaca, Sadettin Emre Eroğlu, Rıza Sarper Ökten, Erdal Birol Bostancı

**Affiliations:** 1Department of Gastroenterology Surgery, Afyonkarahisar State Hospital, 03030 Afyonkarahisar, Türkiye; 2Department of Gastroenterology Surgery, Ankara Bilkent City Hospital, 06800 Ankara, Türkiye; drkadri@gmail.com (M.K.Ç.); otervolkan@gmail.com (V.Ö.); erdalbirol.bostanci@sbu.edu.tr (E.B.B.); 3Department of Radiology, Ankara Bilkent City Hospital, 06800 Ankara, Türkiye; neslihan.karaca@hotmail.com (N.K.); s.e.eroglu@outlook.com (S.E.E.); sarperokten@yahoo.com (R.S.Ö.)

**Keywords:** Cancer Cachexia Index, gastric cancer, postoperative complications, Clavien–Dindo, skeletal muscle, inflammation

## Abstract

**Background/Objective**: Cancer cachexia is a multifactorial syndrome that contributes to adverse surgical outcomes in gastric cancer (GC), yet weight-based criteria often fail to detect subclinical cases. This study aimed to assess the prognostic utility of the Cancer Cachexia Index (CXI) in predicting severe postoperative complications after curative gastrectomy. **Methods**: We retrospectively analyzed 301 patients with GC who underwent curative surgery between January 2020 and October 2023. CXI was calculated as L3 skeletal muscle index × serum albumin/neutrophil-to-lymphocyte ratio (NLR), and patients were stratified into low- and high-CXI groups based on sex-specific medians. Postoperative complications were classified using Clavien–Dindo, with grade ≥ III considered major morbidity. Group comparisons included rates of major complications and hospital stay. **Results**: The low-CXI group had significantly lower muscle mass and albumin levels, higher inflammatory markers, and more T4 tumors. Major complications occurred more frequently in this group (*p* < 0.001). In multivariate logistic regression, low CXI independently predicted severe complications (OR: 2.89; 95% CI: 1.42–5.85; *p* = 0.003), alongside older age and smoking. Receiver operating characteristic (ROC) analysis showed a CXI cut-off of 34.75 yielded high specificity (94.86%) for predicting major morbidity. **Conclusions**: Preoperative CXI is an effective predictor of severe postoperative morbidity in GC patients, outperforming traditional nutritional and inflammatory markers. Incorporation of CXI into routine preoperative assessment may enhance surgical risk stratification and guide perioperative optimization.

## 1. Introduction

Gastric cancer (GC) remains a significant global health burden and ranks among the most commonly diagnosed malignancies of the gastrointestinal tract. In 2020, more than one million new GC cases were reported worldwide, placing it fifth in cancer incidence and fourth in cancer-related mortality, with an estimated 769,000 deaths [1]. Despite advances in diagnosis and treatment, the prognosis of advanced-stage GC remains poor, with median survival rarely exceeding one year [2]. Within Europe, GC accounts for approximately 2.8% of all newly diagnosed malignancies and 4.1% of cancer-related deaths, making it the tenth most common cancer and the seventh leading cause of cancer death in the region [3]. In contrast, the burden is substantially higher in East Asia, particularly in countries such as Japan, South Korea, and China, where incidence and mortality are influenced by environmental and dietary risk factors, Helicobacter pylori infection, and widespread screening practices [4].

Cachexia is a frequent and clinically significant syndrome in patients with GC. It is characterized by systemic inflammation, catabolism, and progressive skeletal muscle loss, and is commonly associated with adverse treatment outcomes [5,6]. While several definitions of cancer cachexia exist, unintentional weight loss remains a central criterion [7,8]. In GC, cachexia affects a substantial proportion of patients—up to 60% in some series—and is typically associated with an average weight loss exceeding 10% [6]. However, the reliance on body weight changes alone may underestimate its prevalence and severity, especially in the presence of fluid retention related to peritoneal metastasis or ascites [9].

The prognostic relevance of cachexia in GC is still under investigation. Some studies have demonstrated a significant association between cachexia and reduced overall survival, particularly in patients receiving systemic therapy [10,11]. However, in those undergoing curative gastrectomy, the predictive value of conventional cachexia definitions appears more limited and variable across disease stages and patient subgroups [12,13].

To overcome the limitations of traditional diagnostic frameworks, the Cancer Cachexia Index (CXI) has been introduced as a composite biomarker incorporating three interrelated domains: skeletal muscle mass, systemic inflammation, and nutritional status. The index is derived from the product of the skeletal muscle index (SMI) and serum albumin, divided by the neutrophil-to-lymphocyte ratio (NLR) [14]. Each of these parameters has been individually associated with prognosis in cancer patients [15,16,17], and their combination into a single index provides a more nuanced evaluation of cachexia-related physiological compromise. The clinical utility of CXI has been demonstrated in several malignancies, including lung, hepatobiliary, and hematologic cancers [15,18,19,20]. However, its prognostic significance in the surgical management of GC, particularly in relation to postoperative morbidity, remains insufficiently characterized.

The objective of this study was to assess whether preoperative CXI could serve as a predictor of postoperative complications in patients undergoing curative gastrectomy for GC. We hypothesized that a lower CXI would be associated with an increased risk of severe postoperative morbidity, thereby offering an additional tool for preoperative risk stratification.

## 2. Materials and Methods

### 2.1. Patients

Patients diagnosed with histologically confirmed gastric cancer (GC) between January 2020 and October 2023 were retrospectively identified from institutional records at the Department of Gastrointestinal Surgery. Initially, 374 patients were screened. The inclusion criteria were (1) age ≥ 18 years, (2) histopathological confirmation of primary GC, and (3) availability of a preoperative abdominal computed tomography (CT) scan performed at our center.

The following exclusion criteria were applied: (1) a history of gastric surgery (*n* = 19), (2) a history of other malignant tumors (*n* = 7), (3) preoperative CT imaging performed at an external institution (*n* = 34), and (4) poor image quality or missing data that prevented accurate CXI calculation (*n* = 13). Patients requiring multivisceral resection were considered ineligible at screening to preserve cohort homogeneity (frequent adjunct HIPEC and non-standard perioperative pathways), and no such patients were enrolled. After these exclusions, a total of 301 patients were included in the final analysis.

Clinical and demographic characteristics, laboratory values, radiological parameters, and pathological findings were obtained from the hospital information system (HIS). The study was conducted in accordance with the principles of the Declaration of Helsinki and was approved by the Clinical Research Ethics Committee of Bilkent City Hospital (Approval No: E2-23-5688; Date: 22 November 2023).

### 2.2. Assessment of SMI, CXI, and Cancer Cachexia

Preoperative abdominal CT images were retrieved from the hospital imaging archive. Skeletal muscle area at the third lumbar vertebra (L3) level was measured using the Syngo MultiModality Workplace platform (Siemens Medical Solutions, Forchheim, Germany) [21]. Measurements were manually delineated with semi-automated tools using a standard HU threshold (–29 to +150) [14,22]. The skeletal muscle index (SMI) was calculated by dividing the cross-sectional muscle area (cm^2^) by the square of the patient’s height (m^2^) [23,24]. All measurements were reviewed and confirmed by consensus among three abdominal radiologists to ensure consistency and reliability. If inter-reader variation for the L3 skeletal muscle area (SMA) exceeded 5%, the final value was determined in a joint consensus session with all three radiologists; measurements were not averaged. Owing to semi-automated delineation using a standard Hounsfield unit threshold (–29 to +150), substantial discrepancies were uncommon and occurred in fewer than 5% of cases.

Preoperative blood samples were obtained following at least 10 h of overnight fasting. The neutrophil-to-lymphocyte ratio (NLR) was calculated by dividing the absolute neutrophil count by the absolute lymphocyte count [25]. Serum albumin levels (g/dL) were extracted from hospital laboratory records. The Cancer Cachexia Index (CXI) was calculated using the following validated formula: CXI = (SMI × Serum Albumin)/NLR [15]. To ensure consistent baseline assessment, all CT and laboratory inputs used for CXI were obtained within 4 weeks prior to definitive treatment. For patients proceeding directly to surgery, measurements were collected within 4 weeks before surgery. For patients receiving neoadjuvant therapy (chemotherapy ± radiotherapy), measurements were collected within 4 weeks before treatment initiation. Post-treatment measurements were not used to compute CXI.

The diagnosis of cancer cachexia was made according to the consensus criteria proposed by Fearon et al., defined by at least one of the following: (1) weight loss > 5% over the previous 6 months; (2) weight loss > 2% in individuals with a body mass index (BMI) < 20 kg/m^2^; or (3) weight loss > 2% in the presence of sarcopenia [26].

### 2.3. Statistical Analysis

All statistical analyses were performed using IBM SPSS Statistics version 26.0 (IBM Corp., Armonk, NY, USA). The normality of continuous variables was assessed using the Kolmogorov–Smirnov test and visual methods including histograms. For continuous variables with normal distribution, comparisons between two groups were made using the independent samples *t*-test, while the Mann–Whitney U test was used for non-normally distributed variables. Categorical variables were compared using the chi-square test or Fisher’s exact test, as appropriate. Non-normally distributed continuous variables (e.g., intraoperative blood loss) were summarized as median (range) and compared using the Mann–Whitney U test, whereas categorical perioperative variables (e.g., intraoperative transfusion) were compared using chi-square or Fisher’s exact tests, as appropriate. Where applicable, odds ratios (OR) with 95% confidence intervals (CI) were reported. To identify predictors of Clavien–Dindo classification grade ≥ 3 complications, univariate and multivariate logistic regression analyses were performed. Covariates with *p* < 0.05 on univariate analysis and those supported by clinical plausibility (age, BMI, SMA/SMI, C-Reactive Protein(CRP), albumin, CXI, smoking, sex, surgical approach, TNM stage, and receipt of neoadjuvant therapy) were considered for entry into the multivariable model. The predictive performance of independent risk factors for Clavien–Dindo grade ≥ 3 complications was further evaluated using receiver operating characteristic (ROC) curve analysis. A two-tailed *p*-value of <0.05 was considered statistically significant. Patients were stratified into low- and high-CXI groups using sex-specific cohort medians; values below the sex-specific median were assigned to the low-CXI group and values at or above the median to the high-CXI group.

## 3. Results

We retrospectively evaluated 374 patients diagnosed with gastric cancer (GC) who underwent surgery between January 2020 and October 2023. Of these, 7 patients with a history of other malignancies and 34 patients whose preoperative CT scans were performed at external institutions were excluded. After skeletal muscle assessment at the third lumbar vertebra level, 13 patients were further excluded due to inadequate CXI calculation (caused by poor imaging quality or unavailable laboratory data). Consequently, 301 patients were included in the study. Patients were stratified into low- and high-Cancer-Cachexia-Index (CXI) groups based on sex-specific median CXI values, resulting in 151 patients in the low-CXI group and 150 patients in the high-CXI group (Figure 1). The sex-specific medians were 45.2 in males (range 10.5–95.8) and 38.6 in females (range 9.2–88.4).

### 3.1. Baseline Characteristics

Table 1 summarizes the baseline demographic and clinical characteristics of the study population. The mean age was significantly higher in the low-CXI group compared to the high-CXI group (64.57 ± 12.90 vs. 60.50 ± 11.66 years, *p* = 0.004). No significant differences were observed between the groups in terms of height (*p* = 0.465), sex distribution (*p* = 0.238), or comorbidities such as diabetes mellitus (*p* = 0.249) and hypertension (*p* = 0.829). Body weight and BMI tended to be lower in the low-CXI group, although these differences did not reach statistical significance (*p* = 0.052 and *p* = 0.079, respectively). The low-CXI group had a higher rate of smoking (*p* = 0.056), but the rates of drinking were about the same (*p* = 0.266). Notably, despite only modest and non-significant differences in BMI between groups, cachexia defined by Fearon’s consensus criteria was significantly more prevalent in the low-CXI group (58.7% vs. 21.2%, *p* < 0.001), underscoring the ability of CXI to identify metabolically cachectic patients who may not meet weight-based thresholds.

### 3.2. Laboratory and Inflammatory Parameters

Laboratory and inflammatory parameters are presented in Table 2. Skeletal muscle area (SMA) and skeletal muscle index (SMI) were significantly lower in the low-CXI group (both *p* = 0.001). Preoperative and postoperative serum albumin levels were significantly lower, while CRP levels were significantly higher in the low-CXI group (all *p* = 0.001). The low-CXI group had significantly lower preoperative and postoperative platelet counts, as well as preoperative lymphocyte counts (*p* = 0.003, *p* = 0.004, and *p* = 0.009, respectively). On the other hand, the low-CXI group had a significantly higher neutrophil-to-lymphocyte ratio (NLR) (*p* = 0.001), reflecting a heightened systemic inflammatory response.

### 3.3. Surgical, Oncologic, and Postoperative Outcomes

Table 3 shows that patients in the low-CXI group had a significantly longer postoperative hospital stay (median 8 (range 3–92) vs. 7 (2–61) days, *p* = 0.001). No significant difference was observed between the groups in terms of neoadjuvant chemotherapy (CT) application rates. In the low-CXI group, 107 patients (71.3%) received neoadjuvant CT, while in the high-CXI group, 118 patients (78.1%) received neoadjuvant CT (*p* = 0.174). While ICU length of stay did not differ significantly between groups (*p* = 0.129), the incidence of major postoperative complications (Clavien–Dindo grade ≥ III) was significantly higher in the low-CXI group (22.7% vs. 9.3%, *p* = 0.002). Minor complications (Clavien–Dindo grade < III) were significantly more frequent in the high-CXI group (11.9% vs. 4.0%, *p* = 0.011). The rate of open surgeries was higher in the low-CXI group (90.7% vs. 79.5%, *p* = 0.006), whereas laparoscopic procedures were more common in the high-CXI group. Intraoperative blood loss was comparable between groups (median (range): low-CXI 187 (115–410) mL vs. high-CXI 163 (80–325) mL; *p* = 0.054). Intraoperative transfusion rates were also similar (low-CXI 21/150 (14.0%) vs. high-CXI 18/151 (11.9%); *p* = 0.71).

Regarding oncologic staging, a significant difference in T-stage distribution was observed (*p* = 0.003). The low-CXI group had more T4 tumors (28.7% vs. 15.9%), while the high-CXI group had more T1 tumors (17.2% vs. 6.0%). However, TNM staging did not differ significantly between the groups (*p* = 0.072). Rates of in-hospital mortality (2.7% vs. 0.7%, *p* = 0.174), overall postoperative morbidity (26.8% vs. 21.3%, *p* = 0.265), and neoadjuvant chemoradiotherapy administration (71.3% vs. 78.1%, *p* = 0.174) were comparable between groups.

### 3.4. Predictors of Major Postoperative Complications

Table 4 presents the univariate and multivariate logistic regression analyses for predictors of major postoperative complications (Clavien–Dindo grade ≥ III). In univariate analysis, older age (*p* = 0.002), lower BMI (*p* = 0.006), lower SMA (*p* = 0.033), higher CRP (*p* = 0.005), lower albumin (*p* = 0.001), lower CXI (*p* = 0.001), smoking (*p* = 0.001), and TNM stage (IV–I, *p* = 0.029) were significant predictors. In the multivariate analysis, age (OR = 1.041, 95% CI: 1.009–1.075, *p* = 0.012), low CXI (OR = 0.979, 95% CI: 0.964–0.994, *p* = 0.005), and smoking (OR = 10.539, 95% CI: 3.321–33.443, *p* = 0.001) remained independently associated with major complications.

### 3.5. Discriminative Ability of CXI and Age for Major Complications

Receiver operating characteristic (ROC) curve analysis for CXI in predicting Clavien–Dindo grade ≥ III complications yielded an area under the curve (AUC) of 0.711, indicating moderate discriminative ability (Figure 2). The optimal CXI cut-off value, determined by the maximum Youden index (0.303), was 34.75, with a sensitivity of 35.42%, specificity of 94.86%, positive predictive value (PPV) of 56.67%, and negative predictive value (NPV) of 88.56%.

The AUC for age as a predictor of major complications was 0.656, with an optimal cut-off of 68 years (Youden index = 0.280), sensitivity of 60.42%, specificity of 67.59%, PPV of 26.13%, and NPV of 90.00% (Figure 3).

These results suggest that preoperative CXI demonstrates superior predictive performance compared to chronological age in identifying gastric cancer patients at high risk for major postoperative complications.

## 4. Discussion

In this retrospective cohort study involving 301 patients undergoing gastrectomy for histologically confirmed gastric cancer (GC), we found that a lower preoperative Cachexia Index (CXI) was significantly associated with major postoperative complications, prolonged hospital stay, and advanced tumor stage. These results support CXI’s role as a practical marker for surgical risk estimation, integrating inflammation, nutritional depletion, and sarcopenia into a unified prognostic tool.

Numerous studies have underscored the prognostic value of sarcopenia, hypoalbuminemia, and elevated inflammatory markers in patients undergoing GC surgery [13,14,27,28,29]. CXI integrates multiple dimensions of cachexia, including inflammation, nutrition, and muscle mass, into a single quantifiable index that better captures the multifactorial nature of cancer-related wasting. Our findings are consistent with those reported by Gong et al., who identified low CXI as a marker of poor prognosis in GC patients [14]. Sakurai et al. also demonstrated that a lower CXI was associated with more severe postoperative complications, although statistical significance was not reached in their adjusted models [30]. In contrast, our study confirmed low CXI as an independent predictor of Clavien–Dindo grade ≥ III complications through both univariate and multivariate logistic regression analyses.

Notably, low CXI was significantly associated with more advanced T stage (higher frequency of T4), whereas the overall TNM-stage distribution did not differ significantly between groups (*p* = 0.072; Table 3). This aligns with previous evidence suggesting that cancer cachexia is more prevalent in patients with locally advanced or metastatic disease due to increased tumor-induced catabolism [7,14,30]. As disease progresses, systemic inflammation intensifies, leading to accelerated muscle degradation and metabolic dysregulation, thereby driving down CXI scores. These results extend prior work by demonstrating a robust correlation between tumor biology and host metabolic status.

From a surgical outcome perspective, patients in the low-CXI group had significantly greater morbidity and longer postoperative hospitalization. These findings are biologically plausible given that sarcopenic and hypoalbuminemic patients have impaired immune responses, delayed wound healing, and reduced cardiorespiratory reserve [27,28,29,30,31]. The observed association between low CXI and open surgical approaches may reflect both physician preference for minimally invasive surgery in fitter patients and the higher incidence of advanced tumors in the low-CXI group. Our institution is a high-volume tertiary referral center that manages complex gastric cancer cases from across the country. The predominance of open surgery in our cohort may be attributed to the advanced tumor stages frequently encountered in referred patients and to the ongoing debate regarding the long-term oncologic equivalence of minimally invasive surgery in advanced gastric cancer, which is yet to be fully clarified in contemporary guidelines Notably, CXI remained predictive of major complications irrespective of whether patients underwent open or minimally invasive surgery, underscoring its role in reflecting baseline physiological vulnerability.

The predictive power of CXI was further supported by our ROC curve analysis. A cut-off value of 34.75 yielded an area under the curve (AUC) of 0.711 for predicting major complications, with high specificity (94.86%) and negative predictive value (88.56%). These metrics suggest that patients above this threshold are unlikely to develop severe morbidity, making CXI a valuable screening tool to identify individuals who may benefit from prehabilitation or intensive perioperative care. These findings are in line with prior ROC-based evaluations of CXI in other malignancies, such as biliary tract cancer and lymphoma [15,19]. We selected the ROC-based cut-off by Youden’s index to balance sensitivity and specificity; alternative thresholds can be adopted depending on clinical priorities (e.g., prioritizing sensitivity for broader screening), and combined risk scores (e.g., CXI plus age) warrant prospective development beyond the scope of the present analysis. Implementation is feasible in routine practice because all CXI components (L3 CT, serum albumin, and NLR) are part of standard preoperative work-ups; SMI can be measured on existing PACS workstations or freely available software (e.g., ImageJ version 1.53c) with brief staff training and no substantial additional cost.

Our data also reinforce the limitations of traditional cachexia definitions, such as Fearon’s criteria, which rely heavily on overt weight loss and BMI [26]. In our cohort, a substantial proportion of patients with low CXI did not meet weight-based cachexia thresholds, highlighting the utility of incorporating biochemical and imaging markers. The combination of low skeletal muscle mass, hypoalbuminemia, and elevated neutrophil-to-lymphocyte ratio (NLR) in these patients suggest the presence of metabolically active cachexia, even in the absence of anthropometric decline [14,16,17,32,33].

Interestingly, smoking also emerged as an independent predictor of major postoperative complications. Smoking is known to impair tissue oxygenation and host immune defense, thereby increasing the risk of respiratory and surgical site infections [34,35]. This finding aligns with previous cohort studies and underscores the importance of smoking cessation counseling as part of perioperative optimization. Given that both smoking and low CXI are strong, independent predictors, patients exhibiting both risk factors should be considered at extremely high risk and targeted for intensive multimodal prehabilitation, including mandatory smoking cessation programs.

Furthermore, our findings resonate with the growing literature emphasizing the prognostic significance of inflammatory–nutritional ratios in GC. Elevated systemic immune-inflammation index (SII), CRP-to-albumin ratio, and similar biomarkers have been associated with increased surgical morbidity and inferior survival [29,30,31,32,35,36]. Integrating these markers into a composite index like CXI enables clinicians to capture the multifactorial burden of systemic inflammation and malnutrition more effectively.

The negative prognostic impact of muscle mass loss has also been highlighted in systemic treatment settings. Blauwhoff-Buskermolen et al. reported that progressive sarcopenia during chemotherapy was significantly associated with reduced overall survival in patients with metastatic colorectal cancer, underscoring the systemic consequences of progressive muscle wasting in oncologic care [37].

Despite its strengths, our study is not without limitations. The retrospective, single-center design may limit generalizability. Specifically, the exclusion of patients with preoperative imaging from external institutions, while necessary for ensuring measurement consistency and internal validity, may introduce a selection bias. Therefore, future multicenter studies with standardized imaging and measurement protocols are needed to validate our findings in a broader population. Additionally, long-term oncologic outcomes such as disease-free and overall survival were not assessed. Finally, although we standardized CXI timing to pre-treatment windows (within 4 weeks before surgery or prior to neoadjuvant initiation) to avoid therapy-related shifts, we did not capture dynamic changes in CXI during or after treatment, which warrants prospective longitudinal investigation. Nonetheless, the large sample size, objective CT-based muscle quantification, and robust multivariate analysis enhance the clinical validity of our findings.

## 5. Conclusions

The preoperative Cancer Cachexia Index (CXI) independently predicts major postoperative complications in patients undergoing gastrectomy for gastric cancer. By integrating markers of systemic inflammation, nutritional status, and skeletal muscle mass, CXI offers a more comprehensive and sensitive measure of physiological vulnerability than conventional weight- or BMI-based cachexia criteria. Its application in preoperative risk stratification may facilitate early identification of high-risk patients and enable the implementation of targeted perioperative interventions—such as nutritional optimization and prehabilitation strategies—to improve postoperative outcomes. Future prospective studies are warranted to confirm these findings and to evaluate whether CXI-guided interventions can effectively reduce postoperative morbidity and improve long-term survival.

## Figures and Tables

**Figure 1 diagnostics-15-02284-f001:**
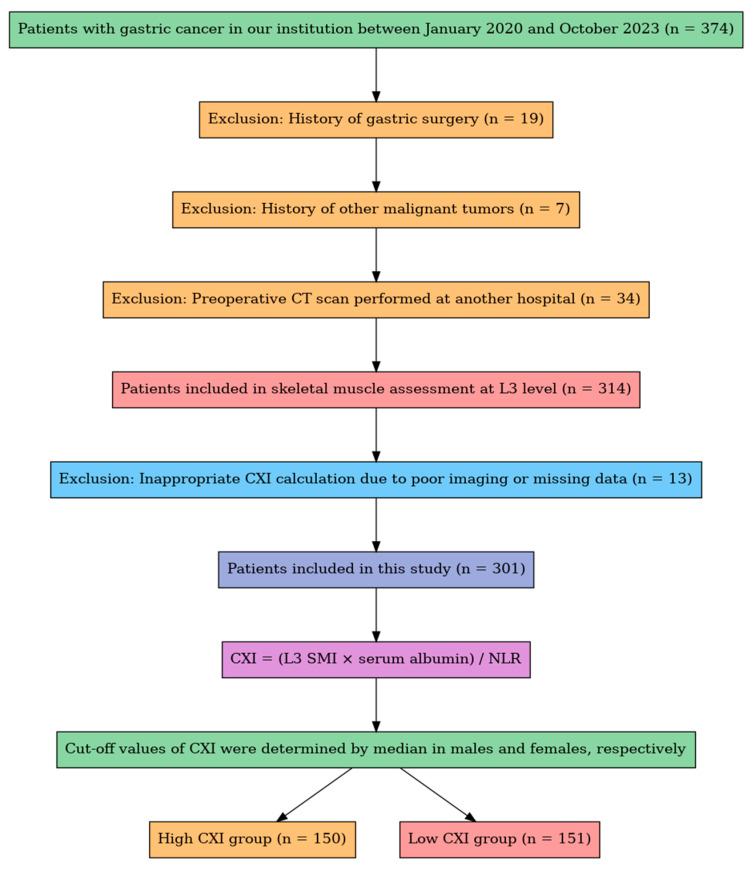
Flowchart of patient selection. A total of 374 patients with gastric cancer were screened. After applying clinical and radiological exclusion criteria, 301 patients were included for CXI calculation and stratified into high- and low-CXI groups based on sex-specific medians.

**Figure 2 diagnostics-15-02284-f002:**
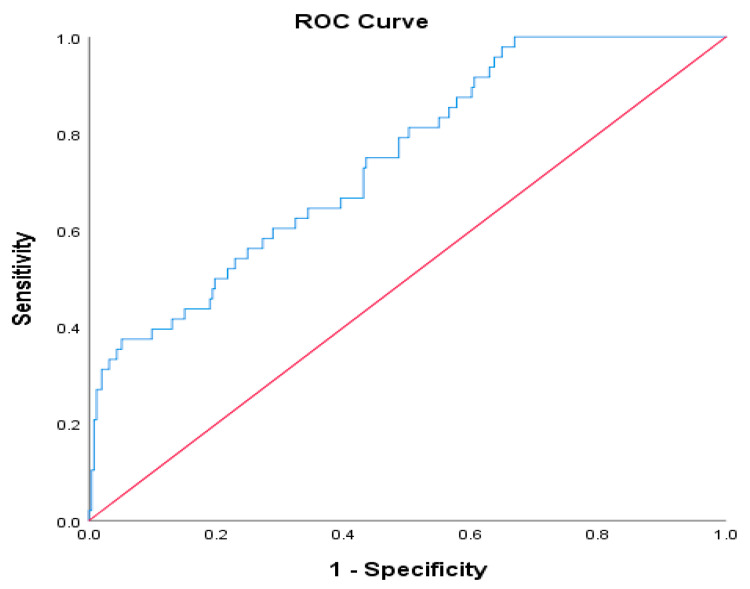
ROC curve for CXI in predicting major postoperative complications (Clavien–Dindo grade ≥ III).

**Figure 3 diagnostics-15-02284-f003:**
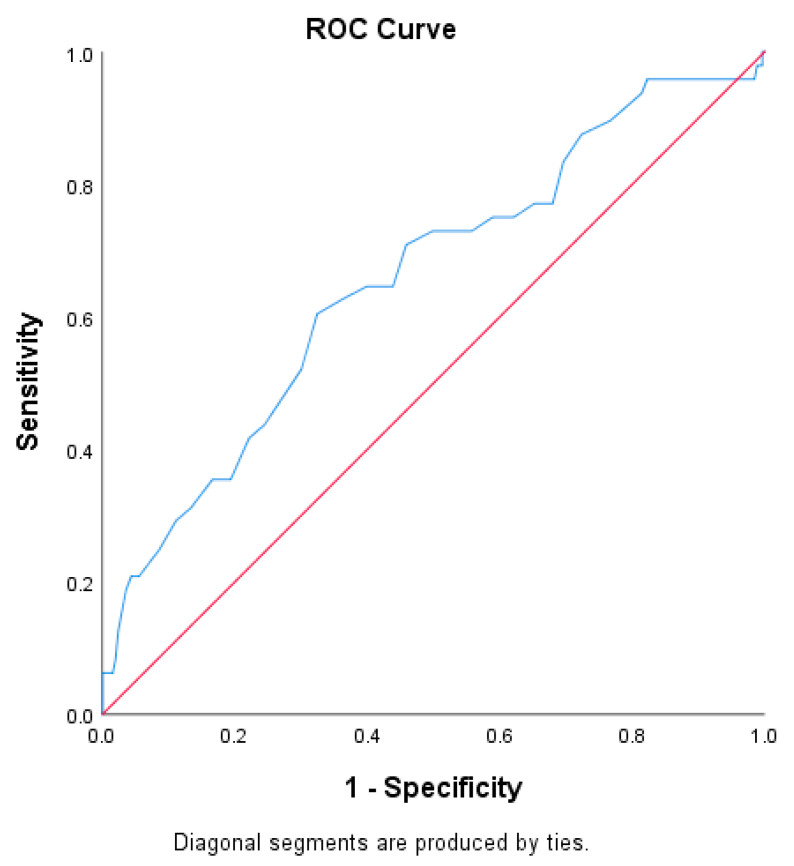
ROC curve for age in predicting major postoperative complications (Clavien–Dindo grade ≥ III).

**Table 1 diagnostics-15-02284-t001:** Comparison of baseline characteristics between low- and high-CXI groups in patients with gastric cancer.

Characteristics		Low CXI (*n* = 150)	High CXI (*n* = 151)	*p*-Value
Age		64.57 ± 12.90	60.50 ± 11.66	**0.004**
Height (cm)		167.77 ± 7.90	168.48 ± 8.75	0.465
Weight (kg)		68.95 ± 13.44	72.23 ± 15.65	**0.052**
BMI (kg/m^2^)		24.44 ± 4.23	25.31 ± 4.36	0.079
Gender	Female	48 (32.0%)	39 (25.8%)	0.238
	Male	102 (68.0%)	112 (74.2%)	
Smoking		88 (58.7%)	72 (47.7%)	0.056
Alcohol consumption		40 (26.7%)	32 (21.2%)	0.266
Hypertension		45 (29.8%)	43 (28.7%)	0.829
Diabetes		17 (11.3%)	24 (15.9%)	0.249
Cachexia (Fearon’s criteria), *n* (%)		88 (58.7%)	32 (21.2%)	<0.001

Data are presented as mean ± standard deviation or number (percentage) unless otherwise indicated. BMI: body mass index; CXI: Cancer Cachexia Index. Differences between groups were assessed using independent *t*-test or chi-square test, as appropriate.

**Table 2 diagnostics-15-02284-t002:** Comparing laboratory and inflammatory factors based on Cachexia Index in people with gastric cancer.

Characteristics	Low CXI (*n* = 150)	High CXI (*n* = 151)	*p*-Value
SMA (cm^2^)	41.75 ± 8.34	48.05 ± 9.94	**0.001**
Preoperative albumin (g/dL)	3.85 ± 0.53	4.19 ± 0.41	**0.001**
Postoperative albumin (g/dL)	2.95 ± 0.37	3.17 ± 0.40	**0.001**
Preoperative DNI	0.10 (0.10–20.40)	0.10 (0.10–9.90)	0.093
Postoperative DNI	0.30 (0.10–12.00)	0.10 (0.10–49.40)	0.617
CRP	9.91 ± 2.98	3.19 ± 0.89	**0.001**
Hemoglobin	12.59 ± 1.42	12.35 ± 1.45	0.154
Preoperative platelet count	260.99 ± 101.42	230.51 ± 74.48	**0.003**
Postoperative platelet count	247.01 ± 92.75	218.59 ± 76.64	**0.004**
Preoperative lymphocyte count	1.38 ± 0.53	2.21 ± 3.82	**0.009**
Postoperative lymphocyte count	0.85 ± 0.51	1.31 ± 4.13	0.177
SMI (cm^2^/m^2^)	41.75 ± 8.34	48.05 ± 9.94	**0.001**
NLR	3.80 ± 2.39	1.73 ± 0.61	**0.001**

Notes: Data are presented as mean ± standard deviation or median (range) for skewed variables. SMA: skeletal muscle area; SMI: skeletal muscle index; NLR: neutrophil-to-lymphocyte ratio; CRP: C-reactive protein; DNI: delta neutrophil index. Statistical comparisons were made using independent *t*-test or Mann–Whitney U test based on data distribution.

**Table 3 diagnostics-15-02284-t003:** Comparing surgical, oncologic, and postoperative outcomes in gastric cancer patients based on their cachexia index.

Characteristics		Low CXI (*n* = 150)	High CXI (*n* = 151)	*p*-Value
Length of hospital stay (day)		8 (3–92)	7 (2–61)	**0.001**
received neoadjuvant CT		107 (71.3%)	118 (78.1%)	0.174
ICU stay (day)		2 (1–74)	2 (1–49)	0.129
Surgical technique	Laparotomy	136 (90.7%)	120 (79.5%)	**0.006**
	Laparoscopy	14 (9.3%)	31 (20.5%)	
T stage	I	9 (6.0%) ^a^	26 (17.2%) ^b^	**0.003**
	II	28 (18.6%) ^a^	24 (15.9%) ^a^	
	III	70 (46.7%) ^a^	77 (51.0%) ^a^	
	IV	43 (28.7%) ^a^	24 (15.9%) ^b^	
TNM Stage	I	24 (16.0%)	43 (28.5%)	0.072
	II	50 (33.3%)	45 (29.8%)	
	III	69 (46.0%)	56 (37.1%)	
	IV	7 (4.7%)	7 (4.6%)	
Mortality Rate		4 (2.7%)	1 (0.7%)	0.174
Morbidity Rate		40 (26.8%)	32 (21.3%)	0.265
Intraoperative blood loss (mL)		187 (115–410)	163 (80–325)	0.054
Intraoperative transfusion, *n* (%)		21/150 (14.0%)	18/151 (11.9%)	0.710
Clavien–Dindo Grade < III		6 (4.0%)	18 (11.9%)	**0.011**
Clavien–Dindo Grade ≥ III		34 (22.7%)	14 (9.3%)	**0.002**

Notes: Data are presented as median (range) or number (percentage). ICU: intensive care unit; T and TNM stages were categorized according to the AJCC 8th edition. ^a,b^ superscripts denote significant pairwise differences within T-stage subgroups (*p* < 0.05). Group differences were evaluated using chi-square test or Mann–Whitney U test as appropriate.

**Table 4 diagnostics-15-02284-t004:** Univariate and multivariate logistic regression analyses of factors linked to major complications after surgery (Clavien–Dindo class ≥ III).

Variables		Univariate Analysis				Multivariate Analysis			
		B	SE	*p*	OR (95% CI)	B	SE	*p*	OR (95% CI)
SMA (cm^2^)		−0.011	0.0054	**0.033**	0.989 (0.978–0.999)	−0.008	0.009	0.362	0.992 (0.975–1.009)
Age		0.048	0.0151	**0.002**	1.049 (1.019–1.081)	0.041	0.016	**0.012**	1.041 (1.009–1.075)
BMI (kg/m^2^)		−0.114	0.0413	**0.006**	0.892 (0.823–0.967)	−0.048	0.058	0.408	0.953 (0.851–1.068)
CXI		−0.021	0.0051	**0.001**	0.979 (0.969–0.989)	−0.021	0.008	**0.005**	0.979 (0.964–0.994)
CRP		0.105	0.0376	**0.005**	1.111 (1.032–1.195)	−0.066	0.061	0.278	0.936 (0.832–1.054)
Surgical Approach (Laparoscopy vs. Open)		−1.550	0.741	**0.037**	0.212 (0.050–0.908)	−0.348	0.833	0.676	0.706 (0.138–3.612)
Gender (Female–Male)		−0.234	0.361	0.516	0.791 (0.390–1.604)	0.742	0.511	0.146	2.100 (0.771–5.719)
Albumin		−1.050	0.293	**0.001**	0.350 (0.197–0.622)	−1.004	0.577	0.082	0.367 (0.118–1.136)
Diabetes		1.430	0.743	0.054	4.192 (0.977–17.980)	−0.760	0.864	0.379	0.468 (0.086–2.544)
Hypertension		0.126	0.353	0.721	1.134 (0.568–2.265)	−0.331	0.417	0.426	0.718 (0.317–1.625)
Smoking		1.890	0.428	**0.001**	6.595 (2.851–15.257)	2.355	0.589	**0.001**	10.539 (3.321–33.443)
Neoadjuvant CT		0.028	0.372	0.940	1.029 (0.496–2.133)	0.748	0.4460	0.093	2.113 (0.882–5.061)
TNM Stage	II–I	0.336	0.748	0.653	1.400 (0.323–6.065)	0.310	0.9770	0.751	1.364 (0.201–9.254)
	III–I	0.521	0.651	0.423	1.684 (0.470–6.030)	−0.851	0.9120	0.351	0.427 (0.071–2.552)
	IV–I	1.443	0.663	**0.029**	4.235 (1.155–15.532)	−0.770	0.963	0.424	0.463 (0.070–3.055)

Notes: Logistic regression analysis was performed to determine factors associated with major postoperative complications (Clavien–Dindo grade ≥ III). SMA: skeletal muscle area; BMI: body mass index; CXI: Cancer Cachexia Index; CRP: C-reactive protein; OR: odds ratio; CI: confidence interval; SE: standard error. Bold *p*-values indicate statistical significance (*p* < 0.05).

## Data Availability

The data presented in this study are available on request from the corresponding author.

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
