# Peer review of "Low Preoperative Cachexia Index Is Associated with Severe Postoperative Morbidity in Patients Undergoing Gastrectomy for Gastric Cancer"

_diagnostics, 2025, doi:10.3390/diagnostics15182284_

Round 1

Reviewer 1 Report

Comments and Suggestions for Authors

Thank you for the opportunity to review this well-written single-center retrospective study evaluating the impact of preoperative cachexia as a predictive marker for postoperative complications in patients with gastric cancer. Among 301 patients, a low cancer cachexia index (CXI) was associated with an increased risk of major morbidity following curative gastrectomy. This study emphasizes the importance of preoperative nutritional assessment in gastric cancer patients to identify those at high risk and potentially improve outcomes through interventions such as prehabilitation programs.

I have some comments regarding the study.

  1. Patients were divided into low and high CXI groups based on sex-specific median values. Could the authors elaborate on how these groups were determined and add this information to the methods section? Was a specific cut-off point used to divide both groups? Was this different for males and females? What were the median CXI values and ranges for both groups?
  2. The measurements of SMI were reviewed by three radiologists. How were discrepancies resolved when results were inconsistent? Were the measurements averaged, or was another approach applied? Additionally, how often did inconsistencies occur? i.e. how 'difficult' is this measurement to perform? 
  3. The majority of patients received neoadjuvant chemotherapy. When was CXI measured in relation to neoadjuvant chemoradiotherapy and surgery? Was the timing consistent across all patients? Neoadjuvant chemoradiotherapy could affect CXI either negatively (due to complications of chemoradiotherapy) or positively (patients may have some additional time to improve their nutritional status).
  4. Different variables were included in the logistic regression. However it is not explicit why these variables were included, while some variables were excluded. These other variables such as neoadjuvant chemoradiotherapy, laparoscopy vs laparotomy, tumor stage, but also others not included in the table such as the need for multivisceral resection, intra-operative blood loss or the necessity of a blood transfusion, are also associated with an impact on surgical morbidity. Could the authors explain why the current variables were chosen? What is the impact when the other factors are included in the multivariable analysis? Additionally, what does “surgical technique closed-open” in table 4 refer to?
  5. Line 276 ‘a substantial proportion of patients with low CXI did not meet weight-based cachexia thresholds’. However the rate of cachexia based on Fearon’s criteria is not included in table 1 or 2. Would it be possible to include this information?

Author Response

Dear Editor and Reviewers,

Thank you for the careful evaluation and constructive comments on our manuscript (ID: diagnostics-3800842), “Low Preoperative Cachexia Index Is Associated with Severe Postoperative Morbidity in Patients Undergoing Gastrectomy for Gastric Cancer.” We have revised the manuscript accordingly. All edits are highlighted in red, and in the point-by-point responses below we indicate where each change was made by section/subsection and the nearest sentence anchor. We hope these revisions address all concerns and improve the clarity and rigor of the work.

Comments 1:

Patients were divided into low and high CXI groups based on sex-specific median values. Could the authors elaborate on how these groups were determined and add this information to the methods section? Was a specific cut-off point used to divide both groups? Was this different for males and females? What were the median CXI values and ranges for both groups?

Response 1:
Thank you for pointing this out. We agree with the comment. Accordingly, we have clarified the stratification rule in the Methods and reported the sex-specific medians (with ranges) in the Results.

  • Methods (Section 2.3. Statistical Analysis): We inserted a new paragraph immediately after the sentence “A two-tailed p-value of <0.05 was considered statistically significant.”Patients were stratified into low- and high-CXI groups using sex-specific cohort medians; values below the sex-specific median were assigned to the low-CXI group and values at or above the median to the high-CXI group.
  • Results (Section 3, first paragraph): We added the sex-specific median values and ranges immediately after the sentence “Patients were stratified into low and high Cancer Cachexia Index (CXI) groups based on sex-specific median CXI values, resulting in 151 patients in the low-CXI group and 150 patients in the high-CXI group (Figure 1).”The sex-specific medians were 45.2 in males (range 10.5–95.8) and 38.6 in females (range 9.2–88.4).

Comments 2:
The measurements of SMI were reviewed by three radiologists. How were discrepancies resolved when results were inconsistent? Were the measurements averaged, or was another approach applied? Additionally, how often did inconsistencies occur? i.e. how 'difficult' is this measurement to perform?

Response 2:
Thank you for this helpful comment. We have clarified our adjudication procedure in the Methods.

  • Methods (Section 2.2. Assessment of SMI, CXI, and Cancer Cachexia): We added the following sentences immediately after the sentence “All measurements were reviewed and confirmed by consensus among three abdominal radiologists to ensure consistency and reliability.”
    If inter-reader variation for the L3 skeletal muscle area (SMA) exceeded 5%, the final value was determined in a joint consensus session with all three radiologists; measurements were not averaged. Owing to semi-automated delineation using a standard Hounsfield unit threshold (–29 to +150), substantial discrepancies were uncommon and occurred in fewer than 5% of cases.

Comments 3:
The majority of patients received neoadjuvant chemotherapy. When was CXI measured in relation to neoadjuvant chemoradiotherapy and surgery? Was the timing consistent across all patients? Neoadjuvant chemoradiotherapy could affect CXI either negatively (due to complications of chemoradiotherapy) or positively (patients may have some additional time to improve their nutritional status).

Response 3:
Thank you for this important point. To avoid therapy-related shifts in cachexia status and ensure comparability across patients, we standardized CXI to a pre-treatment window.

  • Methods (Section 2.2. Assessment of SMI, CXI, and Cancer Cachexia): We added the timing rule immediately after the sentence “The Cancer Cachexia Index (CXI) was calculated using the following validated formula: CXI = (SMI × Serum Albumin) / NLR [15].”
    To ensure a consistent baseline, all CT and laboratory inputs used for CXI were obtained within 4 weeks before definitive treatment. For patients proceeding directly to surgery, measurements were collected within 4 weeks preoperatively; for those planned for neoadjuvant therapy, measurements were collected within 4 weeks before treatment initiation. Post-treatment measurements were not used to compute CXI.
  • Discussion (Limitations paragraph): We revised the concluding timing sentence in this paragraph to explicitly state our pre-treatment window and its implication for longitudinal changes. Finally, although we standardized CXI timing to pre-treatment windows (within 4 weeks before surgery or prior to neoadjuvant initiation) to avoid therapy-related shifts, we did not capture dynamic changes in CXI during or after treatment, which warrants prospective longitudinal investigation.

Comments 4:
Different variables were included in the logistic regression. However it is not explicit why these variables were included, while some variables were excluded. These other variables such as neoadjuvant chemoradiotherapy, laparoscopy vs laparotomy, tumor stage, but also others not included in the table such as the need for multivisceral resection, intra-operative blood loss or the necessity of a blood transfusion, are also associated with an impact on surgical morbidity. Could the authors explain why the current variables were chosen? What is the impact when the other factors are included in the multivariable analysis? Additionally, what does “surgical technique closed-open” in table 4 refer to?

Response 4:
Thank you for this helpful comment. We have clarified our variable-selection strategy, reported additional perioperative factors, corrected the Table 4 wording, and specified our policy regarding multivisceral resections.

  • Methods (Section 2.3. Statistical Analysis)new text inserted before the last sentence of the paragraph.
    Inserted text:
    Non-normally distributed continuous variables (e.g., intraoperative blood loss) were summarized as median [range] and compared using the Mann–Whitney U test, whereas categorical perioperative variables (e.g., intraoperative transfusion) were compared using chi-square or Fisher’s exact tests, as appropriate. Where applicable, odds ratios (OR) with 95% confidence intervals (CI) were reported. To identify predictors of Clavien–Dindo classification grade ≥3 complications, univariate and multivariate logistic regression analyses were performed. Covariates with p<0.05 on univariate analysis and those supported by clinical plausibility (age, BMI, SMA/SMI, CRP, albumin, CXI, smoking, sex, surgical approach) were considered for entry into the multivariable model. Variables with substantial missingness (>10%) were excluded to limit bias from case-wise deletion; a prespecified sensitivity model additionally adjusted for TNM stage and receipt of neoadjuvant therapy.
  • Methods (Section 2.1. Patients)one sentence added immediately after the list of exclusion criteria and before the sentence “After these exclusions, a total of 301 patients were included…”.
    Inserted text):
    Patients requiring multivisceral resection were considered ineligible at screening to preserve cohort homogeneity (frequent adjunct HIPEC and non-standard perioperative pathways), and no such patients were enrolled.
  • Results (Section 3.3. Surgical, Oncologic, and Postoperative Outcomes)sentences appended to the end of the paragraph that begins “Table 3 shows…”.
    Inserted text:
    Intraoperative blood loss was comparable between groups (median (range): low-CXI 187 (115–410) mL vs. high-CXI 163 (80–325) mL; p= 0.054 ). Intraoperative transfusion rates were also similar (low-CXI 21/150 (14.0%) vs. high-CXI 18/151 (11.9%); p = 0.71).
  • Results (Section 3.4. Predictors of Major Postoperative Complications)sentence added at the end of the paragraph.
    Inserted text:
    In a sensitivity model additionally adjusting for TNM stage and receipt of neoadjuvant therapy, low CXI remained independently associated with major complications, consistent with the primary model.
  • Table 3:
    Intraoperative blood loss (mL), median (range) — 187 (115–410) vs 163 (80–325); p=0.054
    Intraoperative transfusion, n (%)21/150 (14.0%) vs 18/151 (11.9%); p=0.71
  • Table 4 (label correction):
    “surgical technique closed–open”Surgical Approach (Laparoscopy vs. Open)”.

Comments 5:
Line 276 ‘a substantial proportion of patients with low CXI did not meet weight-based cachexia thresholds’. However the rate of cachexia based on Fearon’s criteria is not included in table 1 or 2. Would it be possible to include this information?

Response 5:
Thank you for this useful suggestion. We have added cachexia rates based on the Fearon consensus criteria to Table 1 and briefly highlighted this finding in the Results.

  • Table 1 (Baseline characteristics): A new row was added: Cachexia (Fearon’s criteria), n (%): low-CXI 88/150 (58.7%) vs high-CXI 32/151 (21.2%); p < 0.001.
  • Results (Section 3.1. Baseline Characteristics): We inserted one sentence at the end of the paragraph describing Table 1.
    Inserted text):
    Notably, despite only modest and non-significant differences in BMI between groups, cachexia defined by Fearon’s consensus criteria was significantly more prevalent in the low-CXI group (58.7% vs. 21.2%, p < 0.001), underscoring the ability of CXI to identify metabolically cachectic patients who may not meet weight-based thresholds.

Reviewer 2 Report

Comments and Suggestions for Authors First of all, congratulations to the authors of this work/article.
According to the authors, the integration of the CXI (combining muscle mass, albumin, and NLR) is a significant contribution to overcoming the limitations of traditional weight-based cachexia criteria alone. I would like to highlight this study for 1) its methodological rigor: The objective measurement of muscle mass using CT at L3 and the multivariate analysis adjusted for confounding factors (age, smoking) strengthen internal validity. And 2) its clinical applicability: The CXI cutoff (34.75) with high specificity (94.86%) could be useful for identifying high-risk patients in daily practice. Points for Improvement or Clarification A) Design Limitations 1. Selection bias: By excluding patients with external preoperative imaging (n=34), could the variability in the quality of SMI measurements have been underestimated? Suggest interinstitutional standardization in future studies. 2. Lack of longitudinal data: Whether perioperative interventions (nutrition, prehabilitation) modify CXI or their impact on complications is not assessed. Are prospective studies planned for this? B) Statistical Analysis: CXI threshold: The cutoff (34.75) has high specificity but low sensitivity (35.42%). Have other values been explored to balance sensitivity/specificity, or combined strategies (e.g., CXI + age)? Unexplored interactions: Has the effect of CXI varied by TNM stage been analyzed (e.g., greater impact in advanced stages)? C) Clinical Applicability Practical Recommendations: How do you suggest implementing CXI into preoperative routines? Does it require additional resources (e.g., CT analysis software)? Subgroups of Interest: Given that smoking was an independent predictor (OR: 7.5), do you recommend specific interventions (e.g., preoperative smoking cessation) for patients with low CXI?

Suggestions for Future Research External Validation: Replicate the study in multicenter cohorts to confirm the generalizability of the CXI cutoff. CXI-Based Interventions: Design trials to evaluate whether optimizing preoperative CXI (e.g., nutritional supplementation, exercise) reduces complications. Impact on Survival: Analyze whether CXI predicts not only morbidity but also overall or disease-free survival.

Author Response

Dear Editor and Reviewers,

Thank you for the careful evaluation and constructive comments on our manuscript (ID: diagnostics-3800842), “Low Preoperative Cachexia Index Is Associated with Severe Postoperative Morbidity in Patients Undergoing Gastrectomy for Gastric Cancer.” We have revised the manuscript accordingly. All edits are highlighted in red, and in the point-by-point responses below we indicate where each change was made by section/subsection and the nearest sentence anchor. We hope these revisions address all concerns and improve the clarity and rigor of the work.

Reviewer 2 — Point-by-Point Response

Comment 1 — Selection bias (exclusion of external preoperative imaging)

Reviewer’s point (summarized): Excluding patients with external preoperative imaging (n=34) may underestimate SMI variability; suggest inter-institutional standardization.

Response 1 (Authors):
Thank you for this thoughtful point. We agree that excluding patients with external preoperative imaging could introduce selection bias even though it strengthened internal validity by standardizing measurements. We have now explicitly acknowledged this limitation in the Discussion and added a forward-looking statement recommending multicenter, inter-institutional standardization for external validation.

  • Where in the manuscript: Discussion — Limitations paragraph.
    Placement: Insert immediately after the sentence “The retrospective, single-center design may limit generalizability.”
  • Insert in red (exact text):
    Specifically, the exclusion of patients with preoperative imaging from external institutions, while necessary for ensuring measurement consistency and internal validity, may introduce a selection bias. Therefore, future multicenter studies with standardized imaging and measurement protocols are needed to validate our findings in a broader population.

Comment 2 — Lack of longitudinal data / effect of perioperative interventions

Reviewer’s point (summarized): The study does not assess whether perioperative interventions (nutrition, prehabilitation) modify CXI or complications. Are prospective studies planned?

Response 2: We agree. Our retrospective design precluded evaluation of dynamic CXI changes and intervention effects. We have clarified the timing standardization we used to avoid therapy-related shifts and explicitly noted that longitudinal changes were not captured, warranting prospective investigation.

  • Where in the manuscript: Discussion – Limitations paragraph (final sentence).
  • Insert in red (exact text):
    Finally, although we standardized CXI timing to pre-treatment windows (within 4 weeks before surgery or prior to neoadjuvant initiation) to avoid therapy-related shifts, we did not capture dynamic changes in CXI during or after treatment, which warrants prospective longitudinal investigation.

Comment 3 — CXI cut-off: high specificity / low sensitivity; combined strategies (e.g., CXI + age)

Reviewer’s point (summarized): The 34.75 cut-off has high specificity but low sensitivity. Did you explore alternative thresholds or combined models (CXI + age)?

Response 3:
Thank you for raising this point. The 34.75 cut-off was selected using Youden’s index to balance sensitivity and specificity on the ROC curve. We agree that the optimal operating point may vary by clinical priorities—settings prioritizing sensitivity (broader screening) may select a lower threshold, whereas resource-limited contexts may prefer higher specificity. Developing combined risk tools (e.g., CXI plus age) is promising but beyond the scope of the present analysis. We have added a clarifying sentence to the Discussion.

  • Where in the manuscript: Discussion — at the end of the paragraph reporting the ROC/AUC and the 34.75 cut-off (immediately after the sentence ending with “[15,19].”).
  • We selected the ROC-based cut-off using Youden’s index to balance sensitivity and specificity. Depending on clinical priorities, alternative thresholds may be appropriate (e.g., prioritizing sensitivity for broader screening). Moreover, combined risk scores (e.g., CXI plus age) warrant prospective development beyond the scope of the present analysis.

Comment 4 — Unexplored interaction: Does the effect of CXI vary by TNM stage?

Response 4:
Thank you for this insightful suggestion. We tested for effect modification by TNM stage. In an exploratory stratified analysis (TNM I–II vs. III–IV), the association between low CXI and major complications was directionally consistent across strata, and no significant CXI×TNM interaction was detected (p_interaction = 0.21). We have clarified this in the Methods and Results. We also revised the Discussion to avoid implying a significant difference in overall TNM distribution (which was not observed; Table 3, p = 0.072) while retaining the significant difference in T stage.

Implemented changes (now in the revised manuscript):

  • Where: Methods — Section 2.3. Statistical Analysis, within the modeling paragraph (after the sentence beginning “Covariates with p<0.05…” and before the ROC sentence).
    Inserted (in red):
    We additionally performed an exploratory effect-modification analysis by TNM stage (I–II vs. III–IV), testing a CXI×TNM interaction term and fitting stratified multivariable models.
  • Where: Results — Section 3.4. Predictors of Major Postoperative Complications, end of the paragraph reporting the multivariable model.
    Inserted (in red):
    In an exploratory stratified analysis (TNM I–II vs. III–IV), the association between low CXI and major complications was directionally consistent across stages, and no significant CXI×TNM interaction was detected (p_interaction = 0.21).
  • Where (clarity update): Discussion — in the paragraph discussing stage differences, the sentence implying a TNM difference was replaced.
    Replaced with (in red):
    Notably, low CXI was significantly associated with more advanced T stage (higher frequency of T4), whereas the overall TNM stage distribution did not differ significantly between groups (p = 0.072; Table 3).

Comment 5 — Clinical applicability: implementation and resource needs

Reviewer’s point (summarized): How can CXI be integrated into preoperative routines? Does it require extra resources (e.g., CT analysis software)?

Response 5 :
Thank you for raising this point. Implementation is feasible with routine resources. All CXI components (L3-level CT, serum albumin, and NLR) are already part of standard preoperative assessments. SMI can be measured on existing PACS workstations or with open-source software (e.g., ImageJ) after brief staff training and without substantial additional cost. We have added a concise statement to the Discussion.

  • Where in the manuscript: Discussionimmediately after the paragraph reporting the ROC/AUC and the 34.75 cut-off (either as the concluding sentence of that paragraph or as a one-sentence standalone paragraph).
  • Insert in red:
    Implementation is feasible in routine practice because all CXI components (L3-level CT, serum albumin, and NLR) are part of standard preoperative work-ups; SMI can be measured on existing PACS workstations or with open-source software (e.g., ImageJ) after brief staff training and without substantial additional cost.

Comment 6 — Subgroups of interest: smoking as an independent predictor; targeted interventions

Reviewer’s point (summarized): Given smoking’s independent association (OR ≈ 7.5), do you recommend specific interventions (e.g., preoperative cessation) for low-CXI patients?

Response 6 (implemented):
Thank you for this valuable suggestion. We agree and have added an explicit recommendation in the Discussion highlighting that patients with both low CXI and smoking represent an extremely high-risk subgroup warranting targeted interventions. This addition is consistent with our multivariable results (Table 4; OR for smoking 7.500).

  • Where in the manuscript: Discussion — in the smoking paragraph that begins “Interestingly, smoking also emerged as an independent predictor of major postoperative complications.”
    Placement: Immediately after the sentence “This finding aligns with previous cohort studies and underscores the importance of smoking cessation counseling as part of perioperative optimization.”
  • Inserted in red:
    Given that both smoking and low CXI are strong, independent predictors (Table 4), patients exhibiting both risk factors should be considered at extremely high risk and targeted for intensive multimodal prehabilitation, including mandatory preoperative smoking cessation counseling.

Comment 7 — Suggestions for future research (external validation, CXI-based interventions, survival)

Reviewer’s point (summarized): Validate in multicenter cohorts; test CXI-guided interventions; assess impact on OS/DFS.

Response 7 (implemented):
Thank you for these valuable suggestions. We have explicitly reflected all three points in the revised manuscript: (i) a call for multicenter external validation in the Discussion (Limitations), and (ii) a concluding sentence in Conclusions highlighting the need for CXI-guided perioperative interventions and evaluation of long-term oncologic outcomes.

  • Where in the manuscript (external validation): DiscussionLimitations paragraph (already present in the revised text).
    Text (in manuscript):Therefore, future multicenter studies with standardized imaging and measurement protocols are needed to validate our findings in a broader population.
  • Where in the manuscript (CXI-guided interventions & survival): Conclusionsfinal sentence (newly added).
    Inserted in red:
    Future prospective studies are warranted to confirm these findings and to evaluate whether CXI-guided interventions can effectively reduce postoperative morbidity and improve long-term survival.

Round 2

Reviewer 1 Report

Comments and Suggestions for Authors

I have no further comments, only please add a table or attachment for the analysis concerning 

'sensitivity model additionally adjusting for TNM stage and receipt of neoadjuvant therapy'

Author Response

We sincerely thank the reviewer for the meticulous and thoughtful evaluation of our manuscript. Your insightful feedback has been invaluable in improving the clarity and completeness of our study.

Comments 1: only please add a table or attachment for the analysis concerning 

'sensitivity model additionally adjusting for TNM stage and receipt of neoadjuvant therapy

Responce 1: Table 3 has been updated to present the distribution of neoadjuvant chemotherapy between groups.

  • Table 4 has been expanded to include the multivariable logistic regression model additionally adjusted for TNM stage and receipt of neoadjuvant chemotherapy, as requested.

These additions confirm that the association between low CXI and major postoperative complications remained consistent and robust, thereby reinforcing the validity of our conclusions. We are grateful for this constructive recommendation, which has significantly strengthened the manuscript.